# Enhancing Gene Co-Expression Network Inference for the Malaria Parasite *Plasmodium falciparum*

**DOI:** 10.3390/genes15060685

**Published:** 2024-05-25

**Authors:** Qi Li, Katrina A. Button-Simons, Mackenzie A. C. Sievert, Elias Chahoud, Gabriel F. Foster, Kaitlynn Meis, Michael T. Ferdig, Tijana Milenković

**Affiliations:** 1Department of Computer Science and Engineering, University of Notre Dame, Notre Dame, IN 46556, USA; 2Eck Institute for Global Health, University of Notre Dame, Notre Dame, IN 46556, USA; 3Lucy Family Institute for Data & Society, University of Notre Dame, Notre Dame, IN 46556, USAferdig.1@nd.edu (M.T.F.); 4Department of Biological Sciences, University of Notre Dame, Notre Dame, IN 46556, USA; 5Department of Preprofessional Studies, University of Notre Dame, Notre Dame, IN 46556, USA

**Keywords:** malaria, *P. falciparum*, gene co-expression networks, network inference, gene function prediction

## Abstract

Background: Malaria results in more than 550,000 deaths each year due to drug resistance in the most lethal *Plasmodium* (*P.*) species *P. falciparum*. A full *P. falciparum* genome was published in 2002, yet 44.6% of its genes have unknown functions. Improving the functional annotation of genes is important for identifying drug targets and understanding the evolution of drug resistance. Results: Genes function by interacting with one another. So, analyzing gene co-expression networks can enhance functional annotations and prioritize genes for wet lab validation. Earlier efforts to build gene co-expression networks in *P. falciparum* have been limited to a single network inference method or gaining biological understanding for only a single gene and its interacting partners. Here, we explore multiple inference methods and aim to systematically predict functional annotations for all *P. falciparum* genes. We evaluate each inferred network based on how well it predicts existing gene–Gene Ontology (GO) term annotations using network clustering and leave-one-out crossvalidation. We assess overlaps of the different networks’ edges (gene co-expression relationships), as well as predicted functional knowledge. The networks’ edges are overall complementary: 47–85% of all edges are unique to each network. In terms of the accuracy of predicting gene functional annotations, all networks yielded relatively high precision (as high as 87% for the network inferred using mutual information), but the highest recall reached was below 15%. All networks having low recall means that none of them capture a large amount of all existing gene–GO term annotations. In fact, their annotation predictions are highly complementary, with the largest pairwise overlap of only 27%. We provide ranked lists of inferred gene–gene interactions and predicted gene–GO term annotations for future use and wet lab validation by the malaria community. Conclusions: The different networks seem to capture different aspects of the *P. falciparum* biology in terms of both inferred interactions and predicted gene functional annotations. Thus, relying on a single network inference method should be avoided when possible. Supplementary data: Attached.

## 1. Introduction

### 1.1. Motivation and Related Work

Malaria is a deadly disease caused by protozoan parasites of the genus *Plasmodium* (*P.*) that are transmitted by the bite of female mosquitoes [1,2,3]. The most deadly malaria species *P. falciparum* causes more than 0.5 million deaths annually, mostly among children under five years old [4,5,6,7]. Sub-Saharan Africa accounts for 79.4% of malaria cases and 87.6% of deaths [5,8]. *P. falciparum* has evolved resistance to all antimalarial drugs, thus making treatment difficult in areas where multidrug resistant parasites are common [9,10,11,12,13]. The *P. falciparum* research community has developed tools to quickly identify mutations that can be used as markers for drug resistance and genes that are under selection. For some drugs, a causal gene that is the main driver of drug resistance has been identified [14]. However, understanding how mutations in that gene confer resistance is a more difficult problem to solve. Unfortunately, 44.6% of the genes in the *P. falciparum* genome have unknown functions. This lack of knowledge of gene function represents a key challenge to a better understanding of how mutations confer drug resistance. In particular, Gene Ontology (GO) annotations that are often used to describe the biology (e.g., biological processes, molecular-level activities, or cellular structures and localization) of genes are deficient for *P. falciparum* [15]. Deriving novel gene–GO term associations would be a valuable contribution to *P. falciparum* gene annotations.

Biological functions that lead to key traits like drug resistance are controlled by many interacting genes. Hence, studying them as complex networks of gene–gene (or protein–protein) interactions presents promising analytical approaches to uncovering important *P. falciparum* biology [16]. Despite pioneering efforts to obtain physical protein–protein interaction (PPI) data for *P. falciparum* [17,18], high-quality data of this type are lacking [19]. This increases the urgency to understand other interaction/network types. Fortunately, a wealth of gene expression data is available for *P. falciparum* [20], from which gene co-expression networks can be constructed. Gene co-expression networks can be powerfully applied because genes that are functionally related (i.e., that are annotated by the same GO terms) are likely to be coexpressed [21]. Consequently, analyses of a gene co-expression network, where nodes are genes and edges are co-expression relationships between genes over different conditions (e.g., time points or drug treatments), provide a valuable tool for identifying novel (i.e., currently unknown) gene–GO term annotations.

Earlier efforts to build gene co-expression networks in *P. falciparum* have been limited to a single network inference method such as mutual information (MI) [22] or absolute value of the Pearson Correlation Coefficient (absPCC) [23,24], thus leaving other potentially powerful gene co-expression network inference approaches unexplored. Prominent examples include a tree-based measure called Random Forest (RF) [25] and Adaptive Lasso (AdaL) [26]. In fact, it was shown in different species (baker’s yeast, brewer’s yeast, *E. coli*, and *Staphylococcus aureus*) that networks resulting from different gene co-expression network inference methods may be able to give insights into different biological questions [27] and capture different types of regulatory interactions [28]. In addition to investigating *P. falciparum* biology utilizing a single network inference method to build gene co-expression networks, previous efforts have asked limited biological questions about the interacting partners or functions of particular genes [20,22,29,30,31]. However, gene co-expression networks developed using multiple inference methods have not yet been developed for *P. falciparum*. It is unknown whether different network inference methods capture different aspects of *P. falciparum* biology and whether they could inform the important task of systematically predicting GO term annotations for all *P. falciparum* genes.

Here, we fill these gaps by constructing multiple co-expression networks using four prominent network inference methods (MI, absPCC, RF, and AdaL), and by evaluating the networks via systematic and comprehensive prediction of gene–GO term associations. Furthermore, we assess the extent to which the networks are complementary or redundant in terms of their edges (i.e., gene co-expression relationships), as well as predicted functional knowledge. Finally, we apply our inferred networks to the study of endocytosis, which is an essential biological process involved in *P. falciparum* drug resistance.

### 1.2. Our Study and Contributions

We analyzed gene expression data consisting of 247 samples corresponding to 247 combinations of drug treatments and time points [20] (Figure 1). We used this particular dataset because we needed a large dataset with perturbations and preferably with drug perturbations. While newer, RNA-seq data might be available, to the best of our knowledge, the datasets are mostly time course data, and there were no large drug perturbation datasets available when we started our study. We applied four network inference methods to this expression data to infer their respective co-expression networks. Each network inference method mathematically assigns a weight representing the strength of a co-expression relationship between a pair of genes. The highest-weighted gene (i.e., node) pairs are kept as edges in the given network. To choose a weight threshold for distinguishing between edges and nonedges in a network, we adopted a prominent network inference framework called ARACNe [32].

We found that the co-expression networks resulting from the different inference methods are overall highly complementary (i.e., nonredundant). Namely, the networks constructed using MI, absPCC, and AdaL, share only 15–53% of the edges; that is, 47–85% of the edges are unique to each of the networks. Only the network resulting from RF is mostly redundant to the networks resulting from the other three inference methods. This indicates that the different network inference methods largely capture unique features of gene co-expression relationships. Given so many unique edges in almost each of the networks, we considered an additional co-expression network that integrates the edges from all of the individual networks; we refer to this network as Consensus.

Consequently, we used gene–GO term annotations (or associations) to assess each co-expression network by leveraging data from GeneDB https://www.genedb.org/ (accessed on 26 March 2019) and PlasmoDB https://plasmodb.org/plasmo/app (accessed on 26 March 2019) databases. We hid a portion of the existing (i.e., ground truth) gene–GO term annotations (or associations), used the remaining nonhidden associations, along with the given network’s structure (or topology), to predict additional gene–GO term annotations, and evaluated how well the predicted annotations matched the hidden ones, all using crossvalidation. The better this match, the more functionally meaningful the given network’s topology, i.e., the better the corresponding network inference method.

Specifically, we used a prominent paradigm for the unsupervised prediction of functional annotations from a network called network clustering. A cluster is a group of genes in a network that are densely connected to each other or have similar topological patterns [33]. A cluster is deemed functionally meaningful if a statistically significantly high number of its genes are annotated by the same GO term. By extension, one can then predict the other genes in this cluster to also be annotated by this GO term. For the comprehensiveness of this analysis, we used two clustering methods (i.e., a cluster affiliation model for big networks—BigCLAM or BC BC [34]—and Markov Clustering—referred to as MCL [35]), as well as multiple parameter values for each clustering method. We used these clustering methods because they are highly prominent [36,37,38,39,40,41], and also, MCL has been shown to perform consistently well in different contexts [39,40,41,42,43].

We clustered each network and predicted gene–GO term annotations from each cluster of the given network. We evaluated a network’s prediction accuracy over all of its clusters via measures of precision and recall, which compare the predicted annotations with the hidden portion of the existing gene–GO term annotations. Precision is the fraction of the predicted annotations that are correct (i.e., that currently exist); recall is the fraction of the existing annotations that are predicted. Generally, there is a trade-off between precision and recall: higher precision typically means lower recall and vice versa. In biomedical applications, precision is typically favored over recall [44,45,46,47,48] because confirmatory wet lab experiments are time-consuming and expensive, and it is often preferred to follow up on a few higher-quality predictions than many lower-quality ones.

In our own analysis, while some of the networks generated precision scores as high as 87%, the highest recall reached was below 15%. The MI method generated the highest precision, and the others also had reasonably high precision scores. All networks had relatively low recall, thus indicating that none capture a large amount of all existing gene–GO term annotations. In fact, we found the different networks’ predictions to be highly complementary to each other (i.e., the maximum prediction overlap over all network pairs was only 27%). Hence, the different networks seem to capture different aspects of the *P. falciparum* biology, as was the case in baker’s yeast [27].

To supplement the limited functional annotation data in the *P. falciparum* genome, we assigned a confidence score to each (existing, i.e., currently known, and novel, i.e., currently unknown) predicted gene–GO term association based on how many networks supported the given prediction. Similarly, we assigned confidence scores to the inferred gene–gene interactions, i.e., co-expression relationships. We have provided the confidence score-ranked lists of functional predictions and interactions for future community use. These lists could be especially useful given the paucity of comprehensive functional annotations, as well as high confidence PPI data for *P. falciparum*.

In addition to validating inferred networks via gene functional prediction, we investigated the connectivity of genes hypothesized to function together in a biological process. That is, we applied our network approach to recently generated lists of endocytosis-related genes in *P. falciparum* [49]. Endocytosis is an essential biological process for trafficking extracellular material to specific organelles in the cell. Extracellular material is first brought into the cell by early endosomes that mature into late endosomes as they are directed to lysosomes or through the trafficking pathways from the Golgi apparatus. The *P. falciparum* proteins Kelch13 (K13) and EPS15 have been localized to the periphery of the cell and suggested to participate in the machinery to generate early endosomes. The *P. falciparum* protein clathrin functions in an atypical role, which is primarily through trafficking pathways of the Golgi apparatus in other apicomplexan organisms with similar functions suggested in *Plasmodium* [50,51]. K13 is a molecular determinant of artemisinin resistance. EPS15 is a close interacting partner of K13, and it typically functions in canonical endocytosis pathways [52]. Clathrin is also the main structural protein in clathrin-dependent endocytosis. Recently, Birnbaum et al. [49] experimentally determined lists of genes that interact via PPIs with K13, EPS15, and clathrin. Their K13 and ESP15 gene lists were devoid of clathrin, thus suggesting that K13 functions in a clathrin-independent endocytosis pathway. To explore these and other proposed interactions, we used our Consensus co-expression network data for an independent systems-level view as a way to generate new hypotheses about this cellular pathway and the mechanism of resistance for artemisinin. For each of the K13, EPS15, and clathrin gene lists, we found that the genes in a given list are statistically significantly and more densely interconnected with each other in the Consensus network than can be expected by chance. This finding supports and extends the claim that these genes are involved in the same biological process of endocytosis. We also validated some additional hypotheses of Birnbaum et al. [49], which further validates our inferred network data while also providing additional novel insights into the roles of K13, EPS15, and clathrin in *P. falciparum* endocytosis pathways.

## 2. Results and Discussion

### 2.1. Description of Overlap between Inferred Networks

We considered four network inference methods to infer respective co-expression networks from gene expression data GSE19468 containing 4374 genes (Section 4.2, also Appendix A). Each network was named based on its inference method, i.e., MI, absPCC, RF, and AdaL. Among these, as presented in Table 1, all four networks have a similar number of nodes ranging between 4082 and 4374. However, the number of edges and thus network density (the percentage of edges that exist out of all possible edges) varied drastically across the four networks. The one with the lowest network density was AdaL (0.09%), followed by MI (0.77%), then RF (9.08%), and finally absPCC (10.19%).

A lower density indicates a more conservative network inference method, i.e., a method that judges fewer co-expressions as strong enough. Two networks, absPCC and RF, exhibited extremely higher densities than typically sparse real-world networks, including PPI and other biological networks [53]. To handle these two methods being unusually nonconservative, i.e., to highlight the most important gene co-expression relationships within absPCC and RF, as is typically done [54], we examined whether we could remove some proportion of the edges without disconnecting many of the nodes from the given network, as such edges could be viewed as redundant. We did this systematically. Namely, for each of the two networks, we kept 1–100% (in increments of 2% between 1% and 50% and increments of 5% between 50% and 100%) most important (highest-weighted) edges among all of the edges in a given network. Then, for each of these thresholds, we calculated the percentage of nodes from the gene expression data that were in the largest connected component of the resulting thresholded network.

We aimed to preserve at least 85% of the genes from the gene expression data in the largest connected component. This corresponds to keeping at least 30% of the most important edges in the absPCC network. So, we considered the threshold value of 30%, along with two additional, arbitrarily chosen higher values, namely 40% and 50%, for further systematic evaluation. We refer to these three absPCC-based subnetworks as absPCC-0.3, absPCC-0.4, and absPCC-0.5, respectively. Also, from the systematic thresholding procedure described above, we found an interesting pattern with the RF network. Namely, keeping as few as 3% of the most important edges already resulted in more than 95% of the genes from the gene expression data being in the largest connected component. This is why we selected this threshold value of 3%, along with two additional, arbitrarily chosen higher values of 5% and 10%. We refer to these three RF-based subnetworks as RF-0.03, RF-0.05, and RF-0.1, respectively. Appendix A further illustrates the rationale for determining the number of edges kept. Therefore, up to this point, we have constructed 10 networks: one inferred using MI, four inferred using absPCC, four inferred using RF, and one inferred using AdaL (Table 1; note that this table contains an additional network, Consensus, which is discussed below).

Given the co-expression networks constructed using the four inference methods, we examined their pairwise edge overlaps to ascertain their complementarity or redundancy. Note that here we used the smallest subnetworks of absPCC and RF (i.e., absPCC-0.3 and RF-0.03, respectively) to ensure that the networks included the fewest redundant edges. As presented in Figure 2, 46–85% of the edges are unique to each of the MI, absPCC-0.3, and AdaL networks. The fourth network, RF-0.03, is mostly redundant to the other three networks, i.e., only 19.13% of its edges are unique to it. Yet, only 730 edges are common to all four co-expression networks. Note that the 730 edges involve 1181 nodes, and they form 454 connected components, the largest one of which has 13 nodes and 12 edges. Also, only 2 of the 730 edges overlap with the PPIs by [17] (adjusted *p* value of 1).

These results indicate that the complementary co-expression networks may be largely capturing different aspects of *P. falciparum* biology. This is exactly what we found in our subsequent analyses discussed below, in which we show that the networks are predicting complementary gene–GO term annotations. This observation is consistent with results from previous related studies on other systems [27,28]. Given such high network complementarity and in the hope of more accurately capturing regulatory relationships between genes, as is sometimes done [28], we constructed a Consensus network by integrating the edges from the complementary co-expression networks. How to best integrate networks is an important and nontrivial research question [55], but this is not the key aspect of our study. So, here, we used a common sense approach of taking a weighted union of all edges in all considered networks. Specifically, because the different network inference methods have different conservativeness levels judging whether an edge is strong enough, i.e., because they possibly weigh the same edge differently, we aligned these levels between the four networks using min-max normalization. That is, for each network, we normalized the edge weights in a given network to the (0, 1] range. Then, for each edge, we summed its normalized edge weights over the four networks. As such, the resulting Consensus network has edge weights in the (0, 4] range; the higher the weight of an edge, the more important the edge is or the more networks are supporting this edge (or both) (Section 4.2.3). Therefore, in total, we considered 11 co-expression networks in this study (Table 1).

### 2.2. Selecting the Best Clustering Parameter Values for the Inferred Networks

We used two clustering methods (i.e., BigCLAM (BC) and MCL) to generate clusters for predicting gene–GO term associations from each of the 11 co-expression networks. For an unbiased and comprehensive evaluation of each network and a fair comparison of the different networks, we tested multiple parameter values for each clustering method in each network. We used three criteria to select (up to) three clustering parameter values: (i) the parameter value that yields the highest precision in crossvalidation; (ii) the parameter value for which the union of all clusters that are significantly enriched in one or more GO terms contains the most of unique genes, i.e., has the largest gene coverage; and (iii) the parameter value for which the union of all clusters that are significantly enriched in one or more GO terms contains the most unique GO terms, i.e., has the largest GO term coverage.

Intuitively, for each combination of a network, clustering method, and clustering parameter value, we obtained a set of clusters. We used leave-one-out crossvalidation to predict gene–GO term associations from a given set of clusters. That is, given a gene *g*, we hide its existing GO term annotations. Then, we examine whether each of the clusters that gene *g* belongs to is statistically significantly enriched (with adjusted *p* value < 0.05) in an existing GO term *j* using the hypergeometric test; we do this for each GO term under consideration. If so, for such a cluster and GO term *j*, we predict gene *g* being annotated by GO term *j*. We iterate the above process over all genes in the gene expression data and calculate precision and recall, along with the gene coverage and GO term coverage (as defined above) of the significantly enriched clusters. The former serves our criterion (i) above, and the latter two serve our criteria (ii) and (iii) above, respectively. We carried out this entire procedure for each cluster set, i.e., each combination of a network, clustering method, and parameter value. Heuristically, these three criteria maximize the accuracy, as well as coverage of the predicted gene–GO term annotations for each combination. Note that the three criteria can share the same clustering parameter value (Section 4.4). We list the selected parameter values and their resulting numbers of clusters in Table 2.

According to the selected parameter values, we found that the three criteria resulted in quite different parameter values, which in turn resulted in different numbers of clusters and different cluster sizes. This stresses the need to test multiple parameter values for a given clustering method.

### 2.3. Validating the Inferred Networks in the Task of Predicting Gene-GO Term Associations

For each of the 11 networks, given two clustering methods and three selected parameter values per clustering method, there are six combinations of a clustering method and parameter value (Table 2). To further simplify the presentation of results, for a given network, we discarded from further consideration any of its considered combinations that had both lower precision and lower recall than another one of the considered combinations for the same network. That is, we continued considering only the best combinations, i.e., those combinations that are superior to all other combinations for the same network with respect to at least one of precision and recall. For details, see Section 4.4 and Appendix A. This resulted in the following combinations for further consideration: for the MI network: MCL-I1.6, BC-425, and MCL-I2.4 (Appendix A); for the AdaL network: BC-550, MCL-I1.24, and MCL-I1.7 (Appendix A); for Consensus: MCL-I1.7 and MCL-I5 (Appendix A); over all four thresholded absPCC networks: absPCC-BC-650, abcPSS-0.5-BC-650, absPCC-0.3-MCL-I2, absPCC-0.3-MCL-I1.5, and absPCC-0.5-MCL-I3.2 (Appendix A); over all thresholded RF networks: RF-0.1-MCL-I1.28, RF-0.05-BC-600, RF-0.03-BC-700, RF-0.03-MCL-I2, and RF-0.03-MCL-I1.28 (Appendix A).

Given these best combinations of a network, clustering method, and clustering parameter value, we present their precision and recall in Figure 3. We can observe that combinations that yielded higher precision also yielded lower recall, and vice versa. This is expected, as there is a trade-off between precision and recall. Precision is typically favored over recall in biomedicine [44,45]. The recall values of all combinations (and thus of all networks) were below 15%, but some of the networks yielded high precision (Figure 3). In particular, the highest precision value over all combinations for MI was 86.7%, i.e., MI yielded a precision of 86.7%. It is followed by absPCC-0.5, with a precision of 77.8%, RF-0.03 with a precision of 74.3%, and AdaL with a precision of 56.2%. In other words, with respect to precision, the four top-performing combinations of a network, clustering method, and parameter value span all four consideed network inference methods. This means that all inference methods successfully capture meaningful biological signals, i.e., existing gene–GO term annotations. On the other hand, the low recall values of all combinations (specifically, recall values of 2.2%, 0.4%, 2.5%, and 1.8% corresponding to the above four precision values of 86.7%, 77.8%, 74.3%, and 56.2%, respectively) indicate that none of the network inference methods, i.e., their resulting networks, capture a large fraction of all existing gene–GO term annotations.

In theory, the low recall values could in part be due to “noise” in the existing gene–GO term annotation data. Namely, existing gene–GO term annotations have been obtained in many different ways (corresponding to different evidence codes), including experimentally, phylogenetically, computationally (e.g., via sequence or structural similarities), from author or curatorial statements (including “no biological data available” for the latter), or automatically. Some of the evidence codes, especially experimental ones, can be trusted more than the others. So, the co-expression networks might capture some of the evidence codes better than others. Unfortunately, according to the current statistics in the Gene Ontology database, only 12.5% of all gene–GO term annotations in the database have been obtained experimentally. Another possible reason for low recall values, i.e., why no network captures all or even most of the existing functional (i.e., GO) knowledge, could simply be that no computational method should be expected to work well in all biological contexts, i.e., capture all possible functional slices of the cell. If the different networks are predicting complementary gene–GO functional annotations, this could be evidence towards this hypothesis; we examine overlaps between the different networks’ predictions in several places later on in this section.

Going back to our results, each of the four network inference methods has a value (because of high precision), but neither one of them is sufficient on its own (because of low recall). This observation aligns with the motivation of inferring the Consensus network in the hope to increase both the precision and recall values of the individual networks.

However, we found that the Consensus network did not perform the best (Figure 3). In fact, it performed worse than all four top-performing combinations involving the four individual networks that Concensus was constructed from. This could be because the four individual networks capture complementary edges, and simply combining their edges (even via the edge weighing scheme that we use) into Consensus does not necessarily mean producing more biologically meaningful clusters than in individual networks. This might especially hold if the clusters formed by the edges in one individual network capture different GO terms (i.e., different biological knowledge) than the clusters formed by the edges in another individual network; we explore the potential complementarity of the biological knowledge captured by the individual networks later on in this section. If this is the case, integrating the edges from the different networks into the Consensus network and then making functional predictions from this network might weaken the biological signal that can be extracted from the individual networks. But using the individual networks to first make their functional predictions, with each individual network resulting in high precision but low recall and then integrating the predictions, should still result in a high precision that is hopefully close to the precision values of the individual networks, but now also with a higher recall than the recall values of the individual networks. Indeed, we verified that this is what happened: when we integrated the functional predictions of the four top-performing (in terms of precision in Figure 3) combinations of a network, clustering method, and parameter value that cover all four network inference methods, the resulting precision was 66% (compared to the individual precision values of 86.7%, 77.8%, 74.3%, and 56.2%), and the resulting recall was 5% (compared to the individual recall values of 2.2%, 0.4%, 2.5%, and 1.8%). Importantly, even though in the task of predicting gene–GO term annotations the Consensus network did not perform better than the individual networks, the Consensus network did successfully capture drug resistance “biology” relevant to endocytosis (Section 2.5).

Next, as mentioned above, we examined whether the four top-performing combinations of a network, clustering method, and parameter value that cover all four network inference methods, plus the Consensus network, yield redundant or complementary predicted gene–GO term associations. That is, here, we analyze the MI-MCL-I2.4, absPCC-0.5-MCL-I1.32, RF-0.03-MCL-I2.8, AdaL-MCL-I1.7, and Consensus-I5 networks. For each pair of these networks, we measured the overlaps of (1) predicted gene–GO term associations, (2) unique genes that participate in the predicted associations, and (3) unique GO terms that participate in the predicted associations. We did all of this with respect to (i) the predicted existing associations, i.e., the true positives (associations that currently exist and are predicted by the networks) (Figure 4), as well as (ii) the novel associations (associations predicted by the networks that do not currently exist) (Figure 5). We quantified the size of an overlap using the Jaccard index, where a lower Jaccard index indicates a lower redundancy, i.e., a higher complementarity.

For true positive predictions, in total, the five considered networks (i.e., combinations) predicted 169 true positive gene–GO term associations, which involve 109 unique genes and 14 unique GO terms. While most of the pairwise overlaps were statistically significant (adjusted *p* values <0.05), all Jaccard indices were low. That is, the lowest and highest Jaccard indices, respectively, were 3.3% and 27.2% for the predicted true positive gene–GO term associations; 4.7% and 37.5% for the unique genes that participate in the predicted associations; and 11.1% and 70% for the unique GO terms that participate in the predicted associations (Figure 4). For novel predictions, in total, the five networks predicted 174 novel gene–GO term associations, which involve 131 unique genes and 26 unique GO terms. About half of the pairwise overlaps were statistically significant (adjusted *p*-values < 0.05), but again, all Jaccard indices were low. That is, the lowest and highest Jaccard indices, respectively, were 0% and 15.4% for the predicted novel gene–GO term associations; 0% and 18.2% for the unique genes that participate in the predicted associations; and 4.5% and 40.9% for the unique GO terms that participate in the predicted associations (Figure 5).

The above results indicate that the predictions (with respect to both true positives and novel predictions) are largely complementary to each other. This observation strengthens our finding that the different network inference methods capture different biological signals. Importantly, despite the Consensus network not performing well compared to the individual networks in terms of precision, it did uncover gene–GO term annotations not found by the other networks.

We conclude this section by qualitatively complementing the quantitative results thus far on the overlap between gene-functional predictions of the different combinations by breaking down the overlaps by biological processes. Here, for comprehensiveness, we went back to all possible combinations. Namely, remember that we deal with the 11 constructed networks, two clustering methods, and up to three clustering parameter values. For each of the 11×2=22 combinations of a network and clustering method, we considered any gene–GO term association predicted by at least one of the three clustering parameters. Then, we took the union of all such predictions over all combinations. Of all of the predictions in this union, we focused on those that were in the ground truth data, i.e., we considered the true positives over all combinations. We did this because true positives can be trusted more than novel predictions. Finally, we measured, for each of the 22 combinations of a network and clustering method, the percentage of all of the true positives from the union that a given combination predicts.

We show these results in Figure 6, which are broken down by individual GO terms that are then grouped into biological process categories based on the GO terms’ semantic similarities (see Section 4.4 for details). This analysis is intended to complement results from Figure 4a on how much true positive predictions of the different combinations overlap by also providing insights into what biological process(es) the overlaps come from. From Figure 6, we found that the overlaps between the different combinations of a network and clustering method correspond mostly to pathogenesis-related GO terms, as well as to some of the GO terms related to cell cycle and transcription/translation. However, for most of the GO terms *not* related to pathogenesis, the different combinations of a network and clustering method yielded at least somewhat complementary results. The fact that most of the combinations capture the pathogenesis-related GO terms well is encouraging, as these biological processes annotate genes known to be involved in *P. falciparum* infection and immune response.

When we “zoom into” the results for the pathogenesis-related GO terms in Figure 6 with an attempt to compare the four network inference methods to each other with respect to these results, we find that the MI and RF overall capture more of the true positive pathogenesis-related gene-GO term associations from the union of all combinations than the absPCC and AdaL. In more detail, for the MI and RF for many of their combinations of a network and clustering method, a given combination captures many (>70%) of all of the true positives from the union, and it does so for the majority of all pathogenesis-related GO terms. For the absPCC and AdaL, the combination(s) involving the BC clustering method capture many of all the true positives only for certain pathogenesis-related GO terms; for the other pathogenesis-related GO terms or the MCL clustering method, the signal is weaker, although still present.

### 2.4. Ranking Predicted Gene–GO Term Associations and Gene–Gene Interactions

Next, we aim to supplement the limited functional annotation data in the *P. falciparum* genome. Again, here, we deal with the 22 combinations of a network and clustering method, with up to three clustering parameter values for each combination. For each of the 22 combinations, for each gene–GO term association predicted by a given combination, and given up to three adjusted *p* values for a given prediction (corresponding to up to three clustering parameter values), we selected the lowest adjusted *p* value for the prediction. Then, we assigned a confidence score to each gene–GO term prediction made by at least one of the 22 combinations of a network and clustering method and provided the list of all associations ranked by their confidence scores. Intuitively, the more combinations that support a predicted gene–GO term association and the more strongly that a given combination supports a prediction (i.e., the lower the corresponding selected adjusted *p* value), the higher the confidence score of the prediction (see Section 4.6). We provide two ranked gene–GO term association lists: one for the existing associations with 1062 such associations (Appendix A) and the other for the novel associations with 28,826 such associations (Appendix A). For the distribution of the confidence scores, see Figure 7.

Similarly, to supplement the limited gene–gene interactions data in the *P. falciparum* genome, we provide a ranked list of gene–gene interactions with their confidence scores (Appendix A). Intuitively, given all the statistically significantly enriched clusters from the 22 combinations of a network and clustering method, we considered all genes from the clusters along with their edges from the corresponding network. Then, we assigned to each edge a confidence score; intuitively, the more clusters that contain a given edge (i.e., one of its end nodes), and the more functionally meaningful a given cluster is (i.e., the more GO terms it is enriched in and the lower the corresponding adjusted *p* values of the enrichments), the higher the confidence score of the edge (see Section 4.6). The ranked list includes 1,018,420 gene–gene interactions. We visualize the distribution of the interaction confidence scores in Appendix A.

### 2.5. Validating the Inferred Networks Using Endocytosis-Related Biological Signatures

Endocytosis is an essential biological process for trafficking extracellular material in the cell to specific organelles. Extracellular material is first brought into the cell in early endosomes that mature into late endosomes as they are directed to lysosomes or through trafficking pathways from the Golgi apparatus. K13 and EPS15 localize to the periphery of the cell and are suggested to contribute to the generation of early endosomes. Clathrin has been shown to function in *Plasmodium* and other apicomplexa in an atypical role, primarily through trafficking pathways of the Golgi apparatus [50,51]. A recent study [49] identified genes that interact via PPIs with each of the K13, ESP15, and clathrin proteins. The K13 and EPS15 gene lists were devoid of clathrin, thus suggesting that K13 functions in a clathrin-independent endocytosis pathway. Our co-expression networks may provide a systems-level perspective on these recently published data, including new hypotheses for further exploration into this cellular pathway and the mechanism of resistance for artemisinin.

For each of the K13, EPS15, and clathrin gene lists, we measured how densely the genes in a given list are connected to each other, as well as to the genes in the other lists in the Consensus network. Note that even though the Consensus network did not perform the best in terms of precision in the task of predicting gene–GO term annotations, it does cover complementary edges from the individual networks. As such, here we aimed to validate the effectiveness of the Consensus network from the endocytosis perspective. We measured the density of connections between the gene members of each list, between the genes that are in the union of each pair of lists, and between the genes that are in the union or all three lists. Note that if a gene belongs to more than one of the K13, EPS15, and clathrin lists, we do not consider such a gene in this analysis.

We found that the genes in the K13 list are the most densely connected to each other, followed by the genes in the union of the K13 and EPS15 lists (K13-EPS15), followed by the genes in the union of the K13 and clathrin lists (K13-clathrin), and finally followed by the genes in the union of all three lists (All-Endocytosis) (Figure 8). Genes within the K13 list are expected to be densely connected to each other in the network, and so are genes in the K13-EPS15 set, because the K13 and EPS15 gene lists together, while at the same time being devoid of clathrin, contribute to the generation of early endosomes [49]. This means that the Consensus network identifies expected pathways and their interacting proteins, which validates the network. The dense interconnectivity between the genes in the K13-clathrin set is a more interesting result because K13 does not directly interact with clathrin, and it would not necessarily be expected that the K13- and clathrin-interacting genes share the same network. The majority of the material imported by the K13-defined endocytosis pathway is host cell hemoglobin, which is eventually transported to the food vacuole—a lysosome-like structure where hemoglobin is degraded. It is plausible to hypothesize that endosomes generated by K13 are integrated into the trafficking network coordinated by clathrin and its interacting proteins through the Golgi, which is a result that would suggest the separate gene functions in related biological pathways, even though they do not directly interact with each other.

Furthermore, the results suggest that the three gene lists have a potential to operate together in the endolysosomal system to bring material into the cell and transport the material to its destination farther in the cell. Our analysis provides deep, testable extensions of the observed PPI data from [49] for how K13, EPS15, and clathrin are functioning as a system beyond what can be learned from only considering directly interacting protein partners.

Interestingly, the genes in the EPS15 list alone, in the clathrin list alone, or in the two lists combined were not statistically significantly interconnected to each other when analyzed separately from the K13 gene list. This fits a scenario of EPS15 and clathrin having pleiotropic functions and operating in multiple pathways. It is likely that the global transcript data used for the networks does not capture the broad spectrum of interacting partners that either protein has throughout the complex *P. falciparum* cell cycle. Analyzing the gene lists together provides greater context to specific functions in the cell cycle when K13 is essential for cell development. Co-expression networks for specific stages of the *Plasmodium* cell cycle could further define the multiple functions that EPS15 and clathrin preform in the cell. The endocytic mechanisms in *P.* falciparum have not been fully elucidated and are an important area of research [56]. The Karczewski et al. [57] study is a significant step in understanding how K13 functions inside the cell, and it suggests that K13 and clathrin perform different functions inside the cell in separate pathways. Our results suggest that the functional pathways for K13 and clathrin are more correlated than would be expected by only the PPIs found in Karczewski et al. [57]. Further molecular experiments will provide greater context for how the two functional pathways are coordinated and influence artemisinin resistance.

## 3. Conclusions

In this study, we constructed gene co-expression networks for *P. falciparum* using four prominent network inference methods. Then, we evaluated the inferred co-expression networks in terms of their ability to predict existing functional knowledge (i.e., gene–GO term associations) through network clustering and leave-one-out crossvalidation. Our results show that the different networks capture complementary gene–gene co-expression relationships (i.e., interactions) and also predict complementary gene–GO term associations. We have provided ranked lists of inferred gene–gene interactions and predicted gene–GO term annotations for potential future use by the malaria community.

Our study indicates that there does not exist a gold standard co-expression network that captures all aspects of the *P. falciparum* biology. We have shown this via an already systematic and comprehensive analysis (e.g., considering multiple network inference methods, as well as clustering methods and their parameter values) of a large (247-sample) *P. falciparum* gene expression dataset. Yet, this dataset is older. It could be interesting to perform as a subject of future work the same analysis on newer data obtained via more modern biotechnologies or to further improve the analysis by, e.g., considering alternative ways of integrating different networks into a Consensus network, additional network clustering methods, supervised rather than unsupervised function prediction (i.e., classification rather than clustering, although the sparsity of GO annotation data might make the former problematic), accounting for the hierarchical structure of the GO by considering hierarchical relationships between the considered GO terms, or considering only wet lab experimental evidence codes rather than all codes when it comes to gene–GO term association data (if there would be sufficient statistical power for function prediction when considering only experimental evidence codes). Importantly, the result of our current analysis (that there does not seem to exist a gold standard co-expression network that captures all aspects of the *P. falciparum* biology) is further strengthened by the fact that it was already shown in other species that gene co-expression networks resulting from different inference methods may be able to give insights into different biological questions [27] and capture different types of regulatory interactions [28].

Thus, relying on a single network inference method should be avoided when possible. In fact, we have demonstrated that the Consensus network, which combines the interactions from the complementary individual co-expression networks, agrees with the biology of the endocytosis-related cellular pathways and could thus yield new hypotheses about the mechanism of resistance for artemisinin.

## 4. Materials and Methods

### 4.1. Data

#### 4.1.1. Gene Expression Data

We used gene expression data (GSE19468) to construct our co-expression networks curated by Hu et al. [20] in 2009 as follows. Drug perturbations were conducted for 29 drugs across 10 experimental groups (each with a no drug control and two–four drugs); see Table 3). Samples were collected at five–ten time points across the *P. falciparum* intraerythrocytic developmental cycle. This experiment resulted in *P. falciparum* transcription profiles for 247 samples. Transcript abundance levels were obtained using a spotted oligonucleotide microarray [58], with 10,416 probes representing the 5363 genes in the PlasmoDB *P. falciparum* genome version 4.4. In this unprocessed dataset, probes are in rows, and samples are in columns. There have been major revisions in subsequent PlasmoDB *P. falciparum* genome versions. So, rather than using the processed transcription profiles directly, we reprocessed the probe level data to better reflect genome updates and improved normalization methods.

#### 4.1.2. Processing Probe-Level Data

To reflect revisions to the *P. falciparum* genome since 2009, nucleotide sequences for the 10,416 probes on the spotted array were obtained from GEO (GPL7493) and were aligned to the PlasmoDB *P. falciparum* genome version 36 using blast+ (v2.6.0) from NCBI. The 9870 probes that aligned the *P. falciparum* transcriptome (PlasmoDB v36) with a perfect match (bit score ≥ 130) and no secondary alignments (secondary bit scores all < 60) were retained in the dataset. These probes aligned to transcripts for 5075 genes. Probes aligned to transcripts for the same gene were averaged, and genes with nonzero values in >80% of samples were retained in the dataset. This data processing introduced missing values into the dataset. After blast mapping, we end up with 4502 genes with gene names updated in our gene expression data (Appendix A).

#### 4.1.3. Imputing Missing Values in the Gene Expression Data

Some of the network inference methods utilized in this study cannot be used when the underlying gene expression data contain missing data, e.g., absPCC; thus, it was necessary to impute these missing values. We tested seven prominent imputation methods designed for gene expression data. Then, we selected the method that performed the best in the expression data used in this study. By “best”, we mean the method that yields the smallest normalized root mean squared error (NRMSE). In particular, $NRMSE=\frac{\sqrt{mean[{\left(\hat{y}-y\right)}^{2}]}}{std(y)}$, where $\hat{y}$ represents the imputed values and *y* represents the actual values. The seven prominent imputation methods are explained as follows:Multiple imputations by chained equations (MICE) [59] imputes a column (i.e., sample) by modeling each sample with missing values as a function of other samples in a round-robin fashion. That is, given a sample column of interest, namely, *y*, and all other sample columns, namely *X*, a regressor is then fitted on *X* and *y* by learning a regression model from known values in *X* and *y* to predict the missing values in *y*.SVDimpute [60] is a singular vector decomposition (SVD)-based imputation method. Intuitively, a matrix can be recovered asymptotically by only using the significant eigenvalues. That is, given a gene *u*, a regression model of gene *u*- and *k*-most-significant eigenvalues (i.e., eigengenes) is fitted. Then, the learned coefficients of the linear combination of the *k* eigengenes are used to impute the missing values of gene *u*. The processes are repeated iteratively until all missing values are imputed.KNNimpute [60] imputes missing values as follows. First, given a gene *u* with a missing value in sample *j*, *k* other genes without a missing value in sample *j* that are most similar to gene *u* are selected. Then, the weighted average expression level of the *k* selected genes in sample *j* is treated as an estimated expression level for gene *u* in sample *j*, where the weight is the expression similarity (measured using Euclidean distance) of a gene (i.e., among the *k* selected genes) to the gene *u*. We vary *k* from 1 to 24 with an increment of 2.Local least squares imputation (LLSimpute) [61] imputes missing values as follows. First, *k* other genes that are most similar to (i.e., have the largest absolute Pearson correlation coefficients with) gene *u* of interest are selected. It differs from KNNimpute (i.e., *k* is predefined) in that the *k* value for LLSimpute is introduced automatically.SoftImpute [62] imputes missing values by guessing values repeatedly. Specifically, the missing values in the gene expression data are initially filled as zero. Then, a guessed matrix is updated repeatedly by using the soft threshold SVD with different regularization parameters. If the smallest of the guessed singular values is less than the regularization parameter, then the desired guess is obtained. Please refer to [62] for methodological detail.BiScaler [63] was proposed based on SoftImpute but using alternating minimization algorithms. It introduced the quadratic regularization to shrink higher-order components more than the lower-order components such that it offers a better convergence compared to SoftImpute. Please refer to [63] for methodological detail.NuclearNormMinimization [64] imputes missing values by solving a simple convex optimization problem. That is, for a matrix *M* based on a theory that the missing values can be recovered if the number of missing values *m* obeys $m\geq cN^{1.2}rlogN$, where *N* is the number of rows in matrix *M*, *c* is a positive numerical constant, and *r* is the rank of *M*. This algorithm usually works well on smaller matrices.

We used the library “pcaMethods” (version 1.96.0) from Bioconductor R package [65] to perform LLSimpute and the python library “fancyimpute” (version 0.0.5) to perform the remaining six imputation methods. We evaluated the performance of each method on our data as follows: (i) We took rows and columns without any missing values from the expression data as our ground truth data. (ii) We randomly removed 3.14% of the values from the ground truth data. We used 3.14% because it is the percentage of missing values in our expression data. We repeated this process five times and obtained five testing gene expression data. (iii) We applied an imputation method to each of the five testing data and compared the imputed matrix with the ground truth matrix. (iv) We selected the method that yielded the smallest average NRMSE across five runs. (v) Finally, we used the selected “best” method to impute the missing values for our entire gene expression data. It turns out that the MICE (with NRMSE = 0.204) was the best imputation method for our expression data.

#### 4.1.4. Accounting for Cyclical Stage Variation

*P. falciparum* has strong cyclical patterns of transcript expression across the intraerythrocytic development cycle (IDC), and these changes in gene expression tend to swamp other sources of transcriptional variation. To control for this strong cyclical variation, we normalized each drug expression time point by its matched control time point. Since these are already log2-normalized expression profiles, we subtracted the control treatment from the experimental treatments for each time point. Specifically, for a given gene when treated by a drug, the *updated* expression level at time point *t* is obtained by using its *original* expression level minus the expression level at its control treatment at time point *t*. The resulting processed gene expression data thus had 4374 genes and 183 (i.e., 247 − 64) expression levels (i.e., no control groups, as these are used to normalize for cyclical variation). The corresponding data are attached as (Appendix A).

#### 4.1.5. Ground Truth GO Term Annotation Data

We used GO term annotations as our ground truth data to evaluate whether a cluster was statistically significantly enriched in a GO term. GO terms describe the knowledge of the biological domain with respect to three aspects: (1) molecular function, (2) cellular component, and (3) biological process. We focused on biological processes because these terms group genes related to a single objective and are closest to defining gene products involved in the same pathways. We obtained all GO terms that described biological processes and their annotated genes from the two most commonly used databases for *P. falciparum* genes, i.e., GeneDB https://www.genedb.org/ (accessed on 26 March 2019) and PlasmoDB https://plasmodb.org/plasmo/app (accessed on 26 March 2019). Because these two databases cover gene–GO term annotations that are complementary to each other, we combined their annotations and removed duplicates. As such, we obtained 4736 gene–GO term associations that encompass 793 unique GO terms and 2624 unique genes. Note that we used gene–GO term associations obtained via any evidence code. We would ideally restrict only to associations obtained via wet lab experimental evidence codes, but even when considering all evidence codes, the gene–GO term association data are already somewhat limited, especially when further filtering and using only the data that are present in the networks, as discussed below. Also, note that we currently do not explicitly account for relationships between the GO terms in the hierarchical structure of the GO, i.e., we consider a GO term regardless of at what level in the GO hierarchy it resides.

We used the processed data of gene–GO term associations to validate our clusters. Specifically, we treated each GO term as a ground truth cluster such that all genes annotated by this GO term belonged to the same cluster. In general, a valid cluster should include at least three genes. So, we further processed the ground truth data by only keeping those GO terms that annotated at least three genes in the expression data. We denoted such GO terms as relevant GO terms and those associations involved with relevant GO terms as relevant gene–GO term associations. In our experiments, if we mention GO terms, we mean relevant GO terms. We summarize the statistics of our ground truth data in Table 4. The corresponding data are attached as (Appendix A).

#### 4.1.6. Endocytosis Data

PPI data used for the investigation of endocytosis-related genes in the Consensus network were generated recently by Birnbaum et al. [49] in 2020. The study used a BioQ-ID method coupled with mass spectroscopy using asynchronous cultures to define PPIs of three proteins either directly implicated in artemisinin resistance (K13) or as controls (EPS15 and clathrin) to investigate the molecular function of resistance genes fully. Initially, the PPIs of K13 (i.e., the molecular marker for artemisinin resistance) were identified, and a subset was confirmed by cellular localization using immunofluorescence. The *P. falciparum* homolog of EPS15 was identified and confirmed as an interacting partner of K13 [52]. As such, EPS15 was chosen as a positive control for the BioQ-ID experiment, and the genes identified as the PPIs of EPS15 largely overlapped with genes of the K13 PPI list. This added further evidence *K13* was involved in endocytosis. Our study further defined a PPI list for clathrin because it is the main structural protein for canonical endocytosis pathways. The genes of the PPI list for clathrin did not overlap with genes of the K13 or EPS15 PPI lists, thus leading to the conclusion that K13 functions in an endocytosis pathway devoid of clathrin. Our analysis utilized the 173 endocytosis-related genes from [49]. The K13 contains 63 proteins, 60 of which were present in our gene expression data. The EPS15 PPI list contains 49 proteins, 48 of which were in the gene expression data. The clathrin PPI list contains 61 proteins, 58 of which were present in our gene expression data.

### 4.2. Inference of Gene Co-Expression Networks

Given the processed gene expression data (GSE19468), we applied four network inference methods, i.e., mutual information (MI) [32], absolute Pearson Correlation Coefficient (absPCC) [27], Random Forest (RF) [25], and Adaptive Lasso (AdaL) [26]. Moreover, we constructed a Consensus co-expression network by taking the union of networks inferred using these four methods. In this section, we explain how we inferred these co-expression networks.

#### 4.2.1. Network Construction Using ARACNe Framework

We used the ARACNe framework for three network inference methods: MI, absPCC, and RF. Specifically, MI quantifies the mutual dependence (either linear or nonlinear) of two random variables, and absPCC quantifies the linear correlation between two random variables, see [27] for detail. RF [25] is a tree-based method aiming to recover a network involving *n* genes into *n* subproblems such that each subproblem recovers the regulation relationship of a given gene. We used the Python program “GENIE3” to find the edge weight of each gene pair. For methodological detail, see the original publication [25].

Network construction processes of a network include weighting co-expression relationships for each node pair and finding an appropriate threshold to distinguish between edge and nonedge. However, finding an appropriate weight threshold for real-world networks is challenging [27]. Different weight thresholds lead to different networks and, hence, different network topological structures [66]. Because our goal in this study is not to examine the effect of various weight thresholds but to uncover novel functional knowledge about *P. falciparum* using networks, we chose a well-established network construction framework called ARACNe [32] to construct three networks based on the aforementioned three edge weight strategies. ARACNe was originally proposed to infer gene regulatory networks and has been successfully applied to infer gene co-expression networks [22,67,68,69,70]. ARACNe first calculates an appropriate threshold $I_{0}$ based on a null hypothesis that gene pairs are independent of each other if their mutual information is below $I_{0}$. Then, network construction by ARACNe was conducted by first generating 100 bootstraps from the expression data. In each bootstrap, a certain number of microarray samples (i.e., in our case 183) for all genes were randomly selected with replacement and were permuted. Then, a network using mutual information was constructed and pruned using the precalculated threshold $I_{0}$ and data processing inequality (DPI). In particular, given a triangle subgraph (i.e., genes g1, g2, and g3), DPI removes the edge with the smallest weight in the triangle. Intuitively, this is because, in this example, g1 and g3 are interacting with each other indirectly (i.e., through g2) and hence should be removed. Finally, the final network was obtained by keeping those edges that were detected across 100 bootstraps for a statistically significant amount of times. In other words, only nonrandom gene pairs were kept (i.e., adjusted *p* values <0.05 using false discovery rate (FDR) correction). For methodological detail, please refer to the original publication [32].

We directly applied the ARACNe-AP implemented by Lachmann et al. [71] to construct our MI co-expression network. For absPCC, we set the $I_{0}$ to 0.6 according to [72,73] that two metabolites with Pearson Correlation Coefficients ≥ 0.6 are considered as associated. Then we followed the ARACNe framework to obtain our final absPCC network using 100 bootstraps and the DPI technique. For RF, because the tree-based method includes a pruning step that functioned as a weight threshold, we did not set the threshold $I_{0}$. Instead, we directly adopted the GENIE3 implementation from the original publication [25] and the DPI technique for each of the 100 bootstraps. Finally, we used the FDR correction from the ARACNe framework to obtain the final RF co-expression network.

Because the resulting absPCC and RF networks were still very large, we further set the threshold for these two networks by keeping k% most important edges out of all edges in the network without disconnecting the network (i.e., losing too many nodes in the largest connected component). We varied the *k* value from 1 to 100 and selected three such thresholds for each of the absPCC and RF networks. Thus, from this process, we obtained nine networks, i.e., four absPCC-based co-expression networks, four RF-based co-expression networks, and a MI co-expression network.

#### 4.2.2. Network Construction Using Adaptive Lasso

Graphical Gaussian Models (GGMs) are prominent methods for modeling gene associations based on microarray gene expression data [74,75,76,77]. GGMs are often used to obtain an unbiased estimate of the partial correlation between gene *i* and gene *j* such that the partial correlation coefficient is treated as the edge weight of gene *i* and gene *j* in the resulting network. GGMs suffer from two limitations: (i) they assume that the number of microarray experiments is much larger than the number of genes to ensure that the inversion of the covariance matrix in GGMs can be assessed statistically; and (ii) they calculate partial correlation coefficients for all gene pairs in the expression data without a threshold. However, (i) the real-world gene expression data usually come with a larger number of nodes compared to the number of microarray experiments; and (ii) only those gene pairs with strong correlations (i.e., correlations coefficient greater than a certain threshold) are meaningful. To address such limitations of GGMs, Krämer et al. [26] proposed AdaL, a GGMs-based method that uses Lasso penalty to shrink the small edge weights (i.e., coefficients) to zero by using L1 norm regularization, i.e., Lasso. This way, those edges that are not meaningful will be shrunk to zero, i.e., removed from the resulting network. As such, those gene pairs with strong partial correlations are kept and hence form a network. For methodological details of AdaL, please refer to the original publication [26]. Note that we did not apply ARACNe for AdaL, as it already uses the Lasso penalty to shrink small edge weights to zero. Thus, from this process, we constructed another co-expression network, i.e., AdaL.

We used the R package “parcor” (available from the R repository CRAN) [26] to generate our AdaL network using our gene expression data. The input data are defined by an m×n-dimensional matrix, where *m* is the number of microarray treatments (i.e., 183), and *n* is the number of genes (i.e., 4374). The output is an n×n adjacency matrix of the resulting network.

#### 4.2.3. Construction of the Consensus Network

Co-expression networks that are inferred using different inference methods have complementary edges (Section 2). So, we aimed to examine the properties of a network that would be the union of the four aforementioned co-expression networks, namely the Consensus co-expression network. We constructed the Consensus network using MI, smallest absPCC (absPCC-0.3), smallest RF (RF-0.03), and AdaL through the following procedure:Given that we have four networks with different numbers of edges, we derive the following: network 1 has x1 edges, and network 2 has x2 edges, where x1<x2. We argue that the least important edge in network 1 (ranked as x1) should be equally important as the edges that have rank x1 in network 2. This is because each network is inferred via methods with a well-established thresholding strategy, and a network with more edges intuitively has a loosened thresholding strategy compared to a network with fewer edges.We aim to make sure that the higher the rank value of an edge, the more important the edge is. For example, in network 1, an edge with rank x1 means the edge is the most important edge in network 1. So, we reverse the way we rank edges from step 1.To ensure the above two steps are satisfied when we construct the Consensus network, we first calculate the number of edges in all four networks (i.e., AdaL, MI, absPCC-0.3, and RF-0.03). If network 4 is the largest network with x4 edges, we use x4 as our possible maximum rank such that the most important edge in each of the four networks has the same rank, which is x4.According to step 3, network 1 has edge ranks from x4 to x4−x1, network 2 has edge ranks from x4 to x4−x2, network 3 has edge ranks from x4 to x4−x3, and network 4 has ranks from x4 to 1.After we obtain all raw ranks for the edges of each network, we use min-max normalization to normalize the rank of each edge in each network. That is, we first find the maximum (i.e., max also x4) and minimum (i.e., min also 1) edge rank across the four networks. Then, for a given edge between gene *i* and *j* with weight $w_{ij}$, the normalized rank is $\left(w_{ij}-min\right)/\left(max-min\right)$. The resulting normalized rank of edges across four networks spans from 0 to 1.Finally, for a given edge, we sum the weights from the four networks. The collection of all such edges forms the Consensus network. That is, an edge between gene *i* and *j* in the Consensus network has a weight of $w_{ij}=sum\left(w_{ijl}\right)$ for l=1,2,3,4, where *l* is the network ID. Consequently, the resulting Consensus network has a maximum possible edge weight of 4 and a minimum possible edge weight of 0.

With the Consensus network, in total, we constructed and analyzed 11 co-expression networks in this study.

### 4.3. Clustering Methods

Recall that we tested two prominent network clustering methods (i.e., cluster affiliation model for big networks (BigCLAM, referred to as BC) [34] and Markov Clustering (MCL) [35]) to predict gene–GO term associations. BigCLAM is a soft clustering method (i.e., a node can be assigned to multiple clusters), and MCL is a hard clustering method (i.e., a node can only be assigned to one cluster). We selected one method for each type to reduce the effect of clustering type towards prediction accuracy of gene–GO term associations as much as possible. Also, we tested multiple clustering parameters to give each network the best case of advantage.

#### 4.3.1. BigCLAM

BigCLAM (BC) is a non-negative matrix factorization-based algorithm that assumes the overlaps between communities are densely connected. It also detects densely overlapping and hierarchically nested communities. For methodological detail and implementation, please refer tothe original publication [34]. We applied BigCLAM to each of the 11 networks inferred in this study. Specifically, for a given network, we ran the BigCLAM with different parameters, i.e., the number of resulting clusters. We varied it from 50 to 700 with an increment of 25 from 50 to 500 and an increment of 50 from 500 to 700. Note that the number of resulting clusters is a predefined number, thus meaning that the actual number of clusters can be different from the parameter. Then, we tested the prediction performance of each network_clustering_parameter combination and systematically selected up to three combinations based on three predefined selection criteria; see Section 4.4 for details.

#### 4.3.2. MCL

MCL is an efficient random walk-based algorithm that assumes nodes that are densely connected to each other are similar to each other. MCL has been widely applied to detect the modules in the networks [41,78,79]. It takes the adjacency matrix representation and uses expansion and inflation stochastic processes to make the densely connected area denser and the sparsely connected area sparser. We applied MCL to each of the 11 networks and varied the inflation parameters within the range of 1.2 to 5.0 (i.e., as suggested from the user manual https://micans.org/mcl/ (accessed on 1 June 2020)). A smaller inflation parameter tends to result in a smaller number of clusters with bigger cluster sizes compared to a larger inflation parameter. Specifically, we tested 33 inflation parameters with an increment of 0.02 for inflation from 1.2 to 1.4, an increment of 0.1 from 1.4 to 3, an increment of 0.2 from 3 to 4, and an increment of 1 from 4 to 5 for each network. Then, we tested the prediction performance of each combination of a network, a clustering method, and a clustering parameter, and we systematically selected up to three combinations based on three predefined selection criteria; see Section 4.4 for details.

### 4.4. Predicting and Evaluating Gene–GO Term Associations from Clusters

We established a systematic parameter selection and evaluation framework to evaluate each of the combinations of a network, a clustering method, and a clustering parameter value as follows:For the given gene expression data, we test against relevant GO terms to predict and evaluate the accuracy of predicted gene–GO term associations for each of the combinations.For all clusters from a given combination, we use hypergeometric test (Section 4.4.1) to compute the probability scores (i.e., *p* values) of the enrichment significance between each pair of a given cluster and a GO term. If a cluster is statistically significantly enriched in at least one GO term (i.e., *p*-value < 0.05), we mark this cluster as an enriched cluster. We test all clusters and obtain the significantly enriched clusters.In parallel, we make gene–GO term association predictions using significantly enriched clusters and GO terms via leave-one-out crossvalidation [80] as follows:
First, we hide a gene *i*’s GO term knowledge at a time.Second, we test whether each of the clusters that gene *i* belongs to is significantly enriched in any GO term. If such a cluster is statistically significantly enriched by a GO term *j*, we predict gene *i* annotated by GO term *j*.Third, we repeat the above two steps for every gene that has at least one existing GO term annotation. Then, we use precision and recall to evaluate prediction accuracy. The precision is the percentage of correct predictions among all predictions we make. The recall is the percentage of correct predictions among all existing gene–GO term associations. Because there is always a trade-off between precision and recall, we use precision as our parameter selection criteria (3). We do this because we believe that in biomedicine for wet lab validation of predictions, it is more important to have a high precision if we can not have both high precision and recall [44,45].According to our three selection criteria, each combination has up to three clustering parameter values. Different selection criteria could end up with the same clustering parameter. This is why we have up to three selected clustering parameters for a given combination of a network and a clustering method.Because we also aim to compare prediction performance across different networks, we first select the best clustering parameter based on their leave-one-out crossvalidation precision and recall for each of the 11 networks. Specifically, for two clustering parameters, if parameter 1 has a higher precision and a higher recall or a similar recall compared to parameter 2, we select parameter 1. If parameter 1 has a higher precision but a lower recall compared to parameter 2, we keep both. For those selected “best” parameters of each network, we further compare them using the same selection criteria to find the combinations that yield the best prediction accuracy in terms of precision and recall.We then qualitatively analyze our predicted gene–GO term associations using relevant biological pathways. In particular, we visualize how effectively each of the 22 combinations predicts true gene–GO term associations as a heat map showing the proportion of gene–GO term associations correctly predicted for each GO term (rows) by a given combination (columns). GO terms were grouped using semantic similarity using a web tool called REVIGO [81]. Default REVIGO parameters were used to analyze the list of GO terms. The semantic similarity groupings and descriptions from REVIGO were exported by utilizing the “Export to TSV” option under the TREE MAP view.Finally, we perform deep dive analysis for those selected combinations from step 5 using the Jaccard index and overlap coefficient. Both methods measure the overlaps between two sets. In particular, given set A and set B, the Jaccard index is $\frac{\left|A\cap B\right|}{\left|A\cup B\right|}$, and the overlap coefficient is $\frac{\left|A\cap B\right|}{min(|A|,|B|)}$. The Jaccard index results in more accurate results when the sizes of sets A and B are close to each other, while the overlap coefficient results in more accurate results for small data when the sizes of set A and B are far away from each other. The number of predictions made by different combinations can be very different or similar, which is why we use both indices to quantitatively measure our deep-dive analysis.

#### 4.4.1. Hypergeometric Test

We performed hypergeometric test [82] for each pair of a cluster and a GO term (e.g., cluster 1 and GO term 1). Formally, the background (i.e., population) is the number of genes in the expression data of interest, which is represented as *N*. *C* is a cluster, *G* is a GO term, and *x* is the overlap between the cluster *C* and the GO term *G*; thus, the *p* value is computed as follows:(1)$$P\left(X\geq |x|\right)=1-\sum_{i=0}^{|x|-1}\frac{\left(\begin{array}{c} \left|N\right|\\ i \end{array}\right)\left(\begin{array}{c} \left|N|-|C\right|\\ |G|-i \end{array}\right)}{\left(\begin{array}{c} \left|N\right|\\ \left|G\right| \end{array}\right)}$$

Because we performed multiple tests, we adjusted the *p* values using false discovery rate (FDR) correction as follows:Within each network_cluster_parameter combination, we assume we have c1 clusters and g1 GO terms. We adjust c1×g1 *p* values (i.e., each corresponds to a cluster and a GO term pair). We then use the adjusted *p* values to determine whether a GO term is significantly enriched in a cluster.After we select up to six parameters per network based on our three systematic selection criteria, we recorrect up to 11×6×c1×g1 *p* values (i.e., we have 11 networks) for fair comparison between networks for the given gene expression data. By recorrect, we mean we take the up to 11×6×c1×g1 *p* values and use FDR correction to obtain the adjusted *p* values. We do not correct across all tested clustering parameters because some of the parameters are tested to make sure that we are not missing some of the important clustering parameters. Therefore, adding *p* values from these parameters for test correction can be too conservative and consequently remove lots of true positives.

### 4.5. Assigning Confidence Scores to the Predicted Gene–GO Term Associations

In addition to evaluating the prediction accuracy of different networks, another main goal of this study was to predict and rank gene–GO term associations based on their confidence score. In particular, for a given gene expression data, we provided two lists of ranked gene–GO term associations: (1) existing associations, i.e., true positive predictions (those that come from ground truth data), and (2) novel association predictions (those that do not currently exist), along with their “confidence” scores. Intuitively, the more the networks support an association, and the more significant the association is (i.e., the lower the adjusted *p* value of the association), the more confident the association is. We ranked the associations as follows:Recall that we have 11 co-expression networks and two clustering methods, which totals to 22 combinations of a network and a clustering method (i.e., 22 adjusted *p* values). Recall that we selected up to three clustering parameters per combination, with each corresponding to an adjusted *p* value. We select the smallest adjusted *p* value as the adjusted *p* value for the corresponding combination.We rank predictions based on the number of combinations that support this prediction and their corresponding adjusted *p* values. Specifically, we first take the negative log of the adjusted *p* values (transformed *p* values) such that the smaller the adjusted *p* value a prediction has (i.e., the more important the prediction is), the larger the transformed *p* value is. We then sum the 22 transformed *p* values and obtain one final index for each prediction, i.e., the confidence score. We rank the predictions from high to low based on their confidence scores.

### 4.6. Assigning Confidence Scores to the Predicted Gene–Gene Interactions

Similar to the prediction of gene–GO term associations, we also provided a reference list of gene–gene interactions (GGIs) and their “confidence” scores. Intuitively, the interactions that yield the most meaningful gene–GO term associations are supported by the highest amount of networks and/or are among the top-scoring interactions in their respective network, which will be the most confident interactions. Specifically, the rank of the GGIs can be defined as follows:For each of the 22 combinations, we first identify the statistically significantly enriched clusters and obtain all genes in the clusters, along with their edges from the network.For each identified edge, we assign the negative log-transformed *p* value associated with the cluster that the edge belongs to as its weight. If an edge belongs to multiple clusters, (e.g., genes can be grouped into multiple clusters via BigCLAM), we select the one with the smallest adjusted *p* value (i.e., the largest transformed *p* value).We sum up the 22 transformed *p* values (i.e., each corresponds to a combination) and obtain a final confidence score for each GGI. We rank the GGIs from high to low based on their confidence scores.

### 4.7. Examining the Connectivity of Endocytosis-Related Genes in the Consensus Network

We used the Consensus network to examine the hypotheses of the endocytosis process from Birnbaum et al. [49]. Specifically, if a network shows that the *K13*-related genes and *EPS15*-related genes are close to each other than at random in the network, it would corroborate the initial finding of the Birnbaum et al. study by a complementary analysis on independent data. We would also expect clathrin-related genes to be closer to each other than random and distant from *K13*-related and *EPS15*-related genes with connectivity similar to random in the network. We evaluated the connectivity of each group of endocytosis genes using their network connectivity, i.e., the density, as follows:Given a group of endocytosis genes and assuming *m* genes, we select the induced subgraph (i.e., genes and their interactions) of the genes from a given network, and we calculate the observed density of the subgraph. Network density measures how close a network is to its complete version (i.e., all pairs of nodes are connected).We then randomly select *m* genes from the network and their induced subgraph. We calculate the network density. We repeat this process a thousand times and calculate the z score of the observed density compared to densities from the 1000 random runs.We use 2.0 as the z score threshold to determine whether the observed density is significantly larger than the random densities. In particular, if the z score of a group of endocytosis genes is greater than 2.0, then the group of endocytosis genes is more densely connected than expected by chance.

## Figures and Tables

**Figure 1 genes-15-00685-f001:**
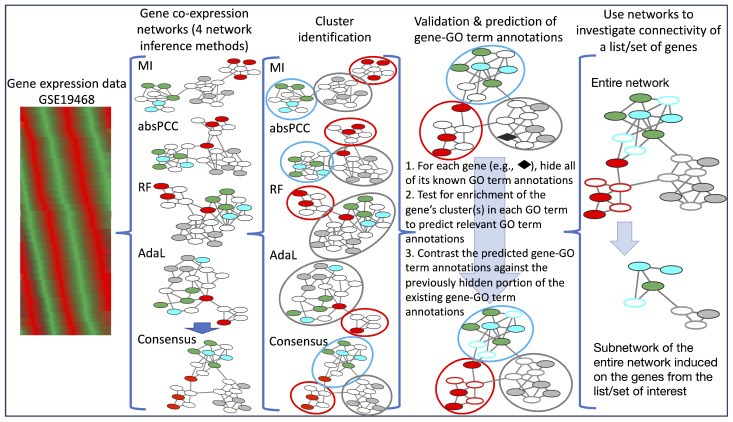
**Summary of our generalizable framework for gene co-expression network construction, validation, and biological application.** We use four network inference methods (MI, absPCC, RF, and AdaL) to construct gene co-expression networks of *P. falciparum*; note that in the illustration, nodes having the same color indicates that they are annotated by the same GO term. We also generate a Consensus network by combining the four individual networks. We use two clustering methods to identify gene clusters’ (hypothesized functional modules) in each network. Then, we use crossvalidation to examine how well each network’s clusters correspond to existing GO terms. In other words, GO term annotations are predicted for all genes whose clusters are statistically significantly enriched in at least one GO term, and the predicted gene–GO term annotations are then contrasted against a previously hidden portion of the existing ones. The Consensus network is used to investigate the connectivity of a gene list/set; in our biological application, we investigate lists of genes that interact with three proteins involved with the biological process of endocytosis.

**Figure 2 genes-15-00685-f002:**
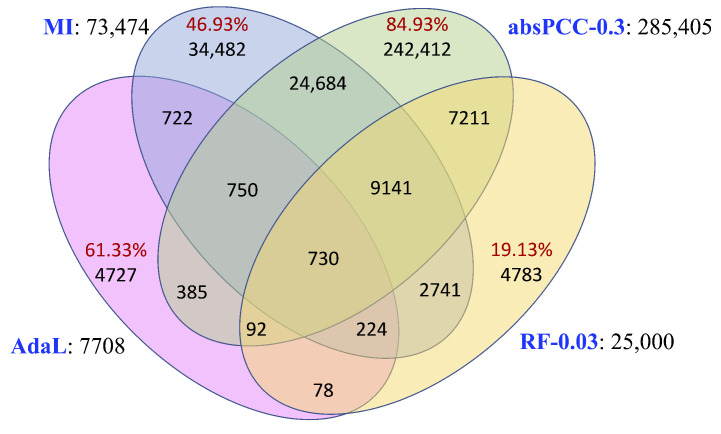
The edge overlaps between the four networks. Each network’s name is colored in blue and is followed by its corresponding number of edges. Within the Venn diagram, each red number is the percentage of all edges in its corresponding network that are unique to the given network. For example, out of all 7708 edges in AdaL, 4727 (i.e., 61.33% of the) edges are unique to AdaL. More detailed information can be found in Appendix A.

**Figure 3 genes-15-00685-f003:**
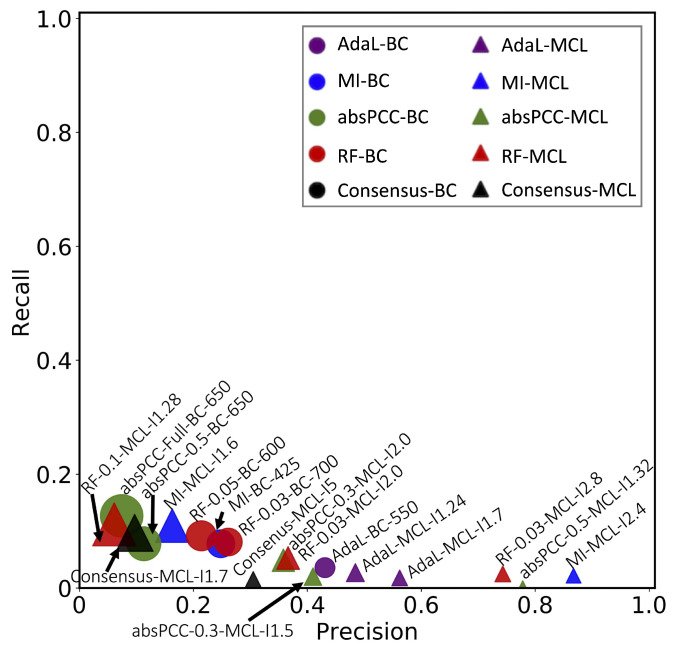
Accuracy of predicting gene–GO term associations using leave-one-out crossvalidation in terms of precision and recall. Each point is a combination of a network, clustering method, and parameter value. The sizes of the points correspond to the number of predictions produced by a given combination. The color of a point corresponds to a network, and the shape of a point corresponds to a clustering method. For example, all purple points correspond to AdaL, of which circles correspond to BC, and triangles correspond to MCL. Note that we use one color (green) for all absPCC networks corresponding to different edge thresholds (e.g., absPCC and absPCC-0.5 share the same color), and we use another color (red) for all RF networks corresponding to different edge thresholds.

**Figure 4 genes-15-00685-f004:**
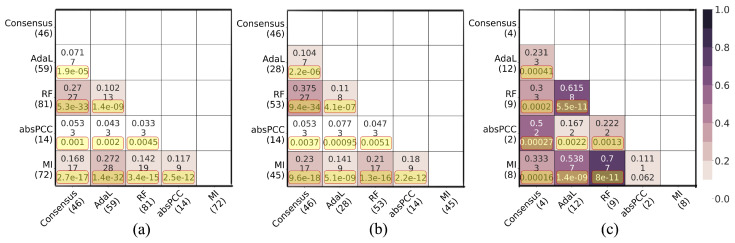
Pairwise overlaps between predictions made by the best combination for each of the five networks with respect to **predicted existing associations, i.e., true positives**. Panel (**a**) shows pairwise overlaps of gene–GO term associations. Panel (**b**) shows pairwise overlaps of unique genes that participate in the predicted gene–GO term associations. Panel (**c**) shows pairwise overlaps of unique GO terms that participate in the predicted gene–GO term associations. Within each panel, the number in parentheses below each network name is the number of corresponding true positives. Within each cell, there are three numbers; the first number is the Jaccard index, the second number is the raw number of the predictions in the given overlap, and the third number is the adjusted *p* value resulting from the hypergeometric test. The yellow boxes highlight the adjusted *p* values that are statistically significant.

**Figure 5 genes-15-00685-f005:**
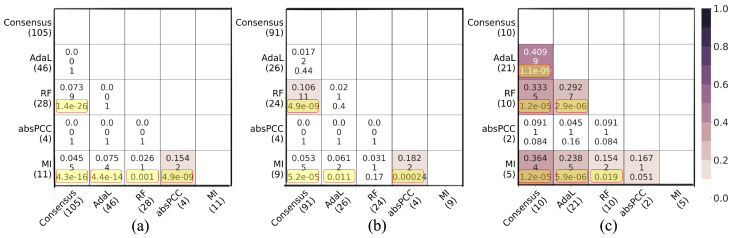
Pairwise overlaps between predictions made by the best combination for each of the five networks with respect to **predicted novel associations, i.e., novel predictions**. Panel (**a**) shows pairwise overlaps of gene–GO term associations. Panel (**b**) shows pairwise overlaps of unique genes that participate in the predicted gene–GO term associations. Panel (**c**) shows pairwise overlaps of unique GO terms that participate in the predicted gene–GO term associations. Within each panel, the number in parentheses below each network name is the number of corresponding novel predictions. Within each cell, there are three numbers; the first number is the Jaccard index, the second number is the raw number of the predictions in the given overlap, and the third number is the adjusted *p* value resulting from the hypergeometric test. The yellow boxes highlight the adjusted *p* values that are statistically significant.

**Figure 6 genes-15-00685-f006:**
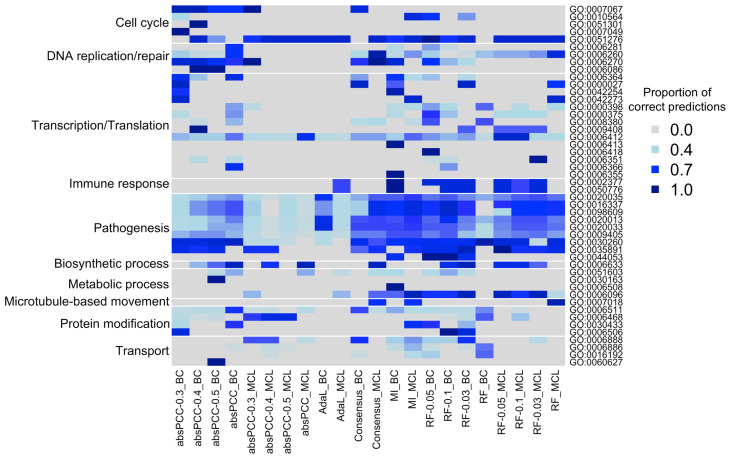
**GO terms and biological process categories captured by the different combinations of a network and clustering method.** Columns correspond to the different combinations. Rows correspond to GO terms that are captured by predicted gene–GO term associations, where the GO terms are then grouped into biological process categories (shown on the left). We visualize the number of gene–GO term associations predicted by a given combination divided by the total number of true positives predicted over all combinations. That is, each cell shows, for a given GO term, the proportion of true positives from the union of all combinations that are predicted by a given combination. The darker blues represent higher proportion values.

**Figure 7 genes-15-00685-f007:**
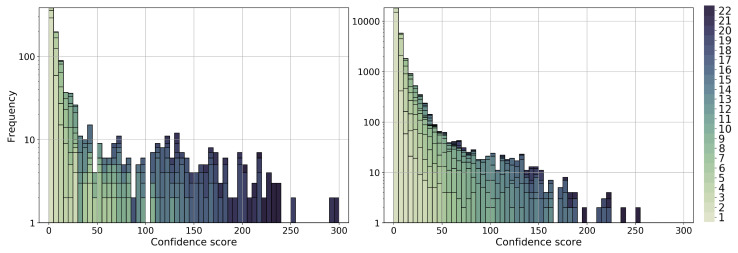
The distribution of confidence scores for predicted gene–GO term associations. The left panel shows the distribution for existing associations, i.e., true positives, and the right panel shows the confidence score distribution for novel associations. The color shades represent the number of combinations of a network and clustering method that support the corresponding association. The darker color the color, the higher the support. Analogous results for gene–gene interactions are shown in Appendix A.

**Figure 8 genes-15-00685-f008:**
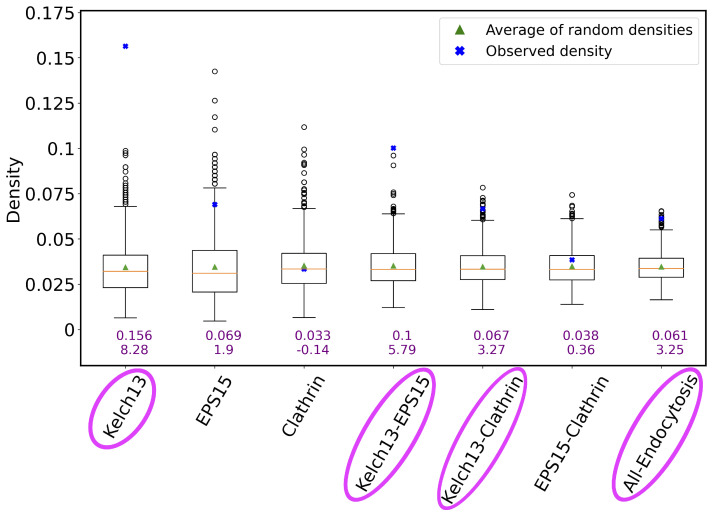
The connectivity of genes in a given list/set (or simply group) compared to the genes’ connectivity expected by chance. The *x* axis shows the considered endocytosis-related gene groups. The *y* axis corresponds to connectivity as measured by network density (for a gene group of size *n*, where there exist *e* edges between the *n* genes, and the density is the ratio of *e* and n2, where the latter is the total possible number of edges between *n* nodes). The boxplot for a given gene group represents the density distribution of 1000 random runs for that group. In particular, if a given gene group has *n* genes, the connectivity by chance is measured by calculating the density of a subnetwork consisting of randomly selected *n* genes and their edges from the Consensus network. The green triangle in each boxplot/for each group is the average density of the 1000 randomly selected subgraphs. The orange line is the median of the 1000 random densities. The blue cross point is the actual (observed) density of the given gene group, whose numerical value is also shown as the top purple number below the given boxplot. The other (bottom) purple number below the given boxplot is the *z* score of the observed density when contrasted against the random densities. The names of endocytosis-related gene groups that are shown in pink ovals correspond to those groups that are significantly more densely connected than at random).

**Table 1 genes-15-00685-t001:** The 11 considered co-expression networks, along with their size and density statistics.

Network	Number of Nodes	Number of Edges	Density
MI	4373	73,474	0.77%
absPCC-0.3	3792	285,405	3.97%
absPCC-0.4	3993	380,539	4.77%
absPCC-0.5	4126	475,674	5.59%
absPCC	4322	951,347	10.19%
RF-0.03	4188	25,000	0.29%
RF-0.05	4323	43,406	0.46%
RF-0.1	4372	86,811	0.91%
RF	4374	868,105	9.08%
AdaL	4082	7708	0.09%
Consensus	4374	333,162	3.48%

**Table 2 genes-15-00685-t002:** The best (i.e., selected) clustering parameter value for each of the three criteria, for each BC and MCL clustering method, and for each co-expression network. The parameter of BC (whose values ranged from 25 to 700) corresponds to the expected number of resulting clusters. The parameter of MCL (that starts with an “I” and whose values ranged from 1.2 to 5) corresponds to the concept of inflation in MCL. Intuitively, the higher the inflation value, the more clusters MCL is expected to return, and hence, the smaller the expected average cluster size is. We list the resulting number of clusters behind each MCL cluster parameter value.

Network	BC Parameter Values			MCL Parameter Values		
	Criterion (i)	Criterion (ii)	Criterion (iii)	Criterion (i)	Criterion (ii)	Criterion (iii)
MI	425	25	450	I2.4 (3775)	I1.6 (133)	I1.7 (571)
absPCC-0.3	350	100	450	I1.5 (138)	I2.9 (908)	I2 (335)
absPCC-0.4	650	125	500	I1.6 (160)	I2.3 (522)	I2 (338)
absPCC-0.5	650	275	475	I1.32 (53)	I2.6 (724)	I2.4 (565)
absPCC	650	225	225	I3.6 (1115)	I2.5 (361)	I3 (711)
RF-0.03	700	25	200	I2.8 (2315)	I1.2 (31)	I2 (1156)
RF-0.05	600	50	350	I3.2 (2985)	I1.24 (19)	I1.9 (830)
RF-0.1	700	50	450	I5 (3953)	I1.28 (10)	I2.1 (1129)
RF	475	50	75	I5 (3519)	I1.7 (6)	I2.5 (201)
AdaL	550	25	75	I1.24 (314)	I1.2 (238)	I1.7 (1105)
Consensus	275	275	200	I5 (3656)	I1.36 (13)	I1.7 (243)

**Table 3 genes-15-00685-t003:** The information about drug treatments and time courses of GSE19468. For example, group 1 has eight time courses for one control group and three drug treatments. That is, there are 8×4=32 gene expression levels for a given gene obtained from group 1. By combining the expression levels across all groups, in total, there are 8×4+5×4+7×5+…+10×3=247 gene expression levels for each of the 5075 genes. Of that amount, 8+5+7+…+10=64 gene expression levels are from control treatments.

Group	Time Points	Treatments
1	8	Control, RoscovitineA, CyclosporineA, FK506
2	5	Control, Colchicine, Na3VO4, StaurosporineA
3	7	Control, ML7, W7, KN93, Staurosporine
4	6	Control, Artemisinin, Chloroquine, Febrifugine, Quinine
5	5	Control, E64, Leupeptine, PMSF, RetinolA
6	6	Control, Apicidin (troph 5 nM), Apicidine (troph IC90)
7	5	Control, Apicidin (schiz IC50), Apicidin (schiz IC90)
8	6	Control, TrichostatinA (IC50), TrichostatinA (IC90)
9	6	Control, Chloroquine (IC50), Chloroquine (IC90), Chloroquine (2 × IC90)
10	10	Control, EGTA (IC50), EGTA (IC90)

**Table 4 genes-15-00685-t004:** Statistics of relevant GO terms and gene–GO term associations.

# of Genes	Number of Samples	Number of Relevant GO Terms	Number of Relevant Gene-0GO Term Associations
4374	183	255	3232

## Data Availability

The datasets generated and analyzed during the current study are available at https://nd.edu/~cone/pfalGCEN/ (Generated on 13 December 2023).

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
