# Peer review of "Enhancing Gene Co-Expression Network Inference for the Malaria Parasite Plasmodium falciparum"

_genes, 2024, doi:10.3390/genes15060685_

Round 1

Reviewer 1 Report

Comments and Suggestions for Authors

Overall this manuscript presents an impressive effort to systematically build, contrast and characterise co-expression networks in P. falciparum, a non-trivial task given lack of well-annotated genome and scarce datasets. The methods are thoroughly explained and the authors have clearly compared a range of methods and set-ups to come forward to their conclusions.

  • Whilst many of the inferred connections were unique to a given method, there were 730 'robust' connections which appeared in the intersection of four networks (Figure 2). A short discussion of these might be interesting, for example are the involved genes part of well defined pathways and are the connections supported by the studies mentioned in the introduction?
  • Could 'noise' in existing GO term annotation databases play a role in the low recall?
  • Would it be worth repeating Figure 3 but ONLY considering the pathogenesis terms? As it seems to be where the main strength of the work lies so might be worth highlighting a bit more.

Author Response

Summary comment: Overall this manuscript presents an impressive effort to systematically build, contrast and characterize co-expression networks in P. falciparum, a non-trivial task given the lack of a well-annotated genome and scarce datasets. The methods are thoroughly explained and the authors have clearly compared a range of methods and set-ups to come forward to their conclusions.

Our response to the summary comment: Thank you for recognizing these positive aspects of our paper.

Comment 1: Whilst many of the inferred connections were unique to a given method, there were 730 'robust' connections which appeared in the intersection of four networks (Figure 2). A short discussion of these might be interesting, for example are the involved genes part of well defined pathways and are the connections supported by the studies mentioned in the introduction?

Our response to comment 1:
This is an excellent question that might be quite hard to answer. A set of edges such as our 730 edges detected by all four network inference methods should ideally be assessed against another, "gold-standard" set of edges. Unfortunately, existing "pathways" linked to specific biological processes are typically gene (i.e., node) sets, not interaction (i.e., edge) sets. The only source of known interactions (i.e., edges) between genes/proteins we could find is the physical protein-protein interaction (PPI) network of P. falciparum by LaCount et al. (Nature, 2005, reference 17 in the paper). So, to address the reviewer's comment, we have contrasted the 730 edges from Fig. 2 against this PPI network, finding only two edges in the overlap (p-value of 1). We have added this information as a brief comment in Section 2.1, when we mention the 730 edges from Fig. 2. Note that as a control, we have also contrasted the top 730 (potentially the most meaningful of all inferred) edges as well as the bottom 730 (potentially the least meaningful of all inferred) edges from our ranked list (Section 2.4)  against the PPI network, resulting in only one and zero edges in the overlap, respectively (with both p-values of 1). Hence, no matter which edge set we have considered, the overlap with the PPI data is almost non-existent. Note that this might not be surprising, as there is no reason to expect that physical bindings between the proteins as captured by PPIs will match gene co-expression relationships. In fact, there have been studies (e.g., by the Trey Ideker lab) that have shown that different types of interactions (e.g., PPIs and genetic interactions) are largely complementary. 

To further address the reviewer's comment, we have analyzed the structure/topology of the 730 edges from Fig. 2. Namely, the 730 edges involve 1,181 nodes, which means that most of the edges in this set are isolated from the other edges in the set. The 730 components form 454 connected components, the largest one of which has 13 nodes and 12 edges, i.e., it is a tree. We have added this information as a brief comment in Section 2.1, when we mention the 730 edges from Fig. 2 in the text. Note that the topologies of the top 730 edges and bottom 730 edges from our ranked list are similar, with more nodes than edges, with hundreds of connected components, and with a quite small largest connected component. 

Comment 2: Could 'noise' in existing GO term annotation databases play a role in the low recall?

Our response to comment 2:
In theory, yes, "noise" in the GO annotation data could in part contribute to low recall. We have added a related discussion as a new (third) paragraph in Section 2.3 of the revised paper. 

Comment 3:
Would it be worth repeating Figure 3 but ONLY considering the pathogenesis terms? As it seems to be where the main strength of the work lies so might be worth highlighting a bit more.

Our response to comment 3:

This might be worth considering, but only if one suspects that the networks should work well only for those GO terms. If instead one is trying to uncover which GO terms a given network works well for, then one should test the networks on all GO terms. And that is what we have done in Fig. 3.

We could in theory run the same analysis as in Fig. 3 but only for pathogenesis-related GO terms. However, given the extremely tight timeline of only five days that we were given by the journal to carry out the revision, in practice, we are unable to do this specific analysis. Performing it would require us to carry out almost all steps of our entire pipeline but for these GO terms only. Namely, we could not just use the existing gene-GO term predictions that we already have from our original analysis, and from these, filter and analyze only the predictions for pathogenesis-related GO terms and compute the resulting precision and recall values. This is because the existing predictions are a result of multiple steps, an early step being correction over all considered GO terms of p-values of the enrichment of each cluster from each network in each GO term, where the p-values determine which gene-GO term predictions are to be made. So, if we were to focus only on pathogenesis-related GO terms, we would need to compute new corrected p-values, which would then mean selecting new best clustering parameters, which would in turn lead to new predictions, etc. So, things would get very complex for the short period of time that we have.

Nonetheless, we believe that even the original paper already addresses the reviewer's comment implicitly in Fig. 6, although not explicitly in terms of precision and recall as in Fig. 3. Namely, as reported in the original paper, Fig. 6 complements the results reported earlier in that section (Section 2.3) by breaking the results down by biological processes, including pathogenesis-related GO terms. According to Fig. 6, as reported in the original paper, "the overlaps between the different combinations of a network and clustering method correspond mostly to pathogenesis-related GO terms, as well as to some of the GO terms related to cell cycle and transcription/translation. However, for most of the GO terms not related to pathogenesis, the different combinations of a network and clustering method yield at least somewhat complementary results."

Moreover, to further address the reviewer's comment, we added a new (last) paragraph to Section 2.3, in which we "zoom into" the results for the pathogenesis-related GO terms in Fig. 6, with an attempt to compare the four network inference methods to each other with respect to these results. In the new paragraph, we discuss our observation that MI and RF overall capture more of the true positive pathogenesis-related gene-GO term associations than absPCC and AdaL, plus some more detailed observations.

Reviewer 2 Report

Comments and Suggestions for Authors

Dear Authors,

I reviewed your manuscript "Enhancing gene co-expression network inference for the malaria parasite Plasmodium falciparum".

It is in-depth, up-to-date research on the chosen subject. The manuscript is well structured, very technical, and thoroughly described.

If I were to make observations, they would mainly refer to the grammatical part: commas, articles, unsplit the infinitive, rewrite sentences for clarity, and so on.

However, I have a significant observation: you describe your study in the present tense as if you were doing it while writing it (e.g., "we analyze" - line 85, "we use" - line 86, "we find' - line 97, and so on). This is not a current practice in writing articles. In general, it is assumed that the research was carried out, and you should describe it, consequently, in the past tense. It doesn't sound very good, as it is! Therefore, I suggest rewriting the manuscript in the past tense!

Comments on the Quality of English Language

My comments on the quality of English are revealed above in the Comments and Suggestions for Authors section. 

Author Response

Summary comment: Dear Authors, I reviewed your manuscript "Enhancing gene co-expression network inference for the malaria parasite Plasmodium falciparum". It is in-depth, up-to-date research on the chosen subject. The manuscript is well structured, very technical, and thoroughly described.

Our response to the summary comment: Thank you for recognizing these positive aspects of our paper.

Comment 1: If I were to make observations, they would mainly refer to the grammatical part: commas, articles, unsplit the infinitive, rewrite sentences for clarity, and so on.

Our response to comment 1: Thank you for the feedback. We have carefully gone through the paper again and made 50+ changes towards improving the readability, such as adding missing commas, adding missing articles, splitting up long sentences into shorter ones, and rewriting parts of sentences. So, we believe that the revised paper reads better than the original one. If the reviewer has any further specific suggestions on what might remain unclear, we would be happy to account for the additional feedback.

Comment 2: However, I have a significant observation: you describe your study in the present tense as if you were doing it while writing it (e.g., "we analyze" - line 85, "we use" - line 86, "we find' - line 97, and so on). This is not a current practice in writing articles. In general, it is assumed that the research was carried out, and you should describe it, consequently, in the past tense. It doesn't sound very good, as it is! Therefore, I suggest rewriting the manuscript in the past tense!

Our response to comment 2:
We believe that this is a subjective preference that depends on one's writing style. The corresponding author of the paper always writes all of her papers in present tense (as in, what are we doing in this study), reserving past tense for discussion of already published work (including what we did in our previous work). This is a common trend among the corresponding author's peers. We acknowledge that we have also seen papers published in past tense. So, we believe that there is no right or wrong choice here. There is only a subjective choice based on one's preferences. And our choice is to use present tense for each current study/paper being reviewed, including this paper. So, we most strongly prefer to keep the paper as is, in present tense.